# Learning Unorthogonalized Matrices for Rotation Estimation

## Abstract

Estimating 3D rotations is a common procedure for 3D computer vision. The accuracy depends heavily on the rotation representation. Rotation matrices, recently, have been popular due to their continuity, especially for pose estimation tasks. The learning process usually incorporates orthogonalizations to generate orthonormal matrices. We observe that common orthogonalization procedures like Gram-Schmidt-based and SVD-based may slow down the training efficiency via a gradient analysis. To this end, we advocate removing orthogonalization from the learning process and learning unorthogonalized 'Pseudo' Rotation Matrices (PRoM). To prove the superiority of PRoM over orthogonalization incorporated methods, we conduct an optimization analysis to explicitly demonstrate that PRoM can converge at a higher rate and to a better solution. By replacing the orthogonalization incorporated representation with our proposed PRoM in various rotation-related tasks, we can achieve state-of-the-art results on large-scale benchmarks.

## 1 Introduction

Estimating 3D rotation is a common procedure in geometry-related tasks such as 3D pose estimation. A 3D rotation is defined by three parameters. Some rotation representations like axis angles or Euler angles are specified by three parameters, while others are over-parameterized, like quaternions (four) and rotation matrices (nine). Overparameterized rotation representations must fulfill certain constraints; for example, matrices $\mathbf{R}$ in rotation groups $SO(n)$ must be orthogonal.

Accurately estimating rotation parameters can be challenging; one reason is that the representation (axis-angles, Euler angles, or quaternions) is discontinuous (Grassia, 1998; Saxena et al., 2009). Recently, (Zhou et al., 2019) proposed a continuous 6-parameter representation for 3D rotations. The so-called '6D representation' simply drops the last column vector of the rotation matrix; the full matrix can be recovered via a Gram-Schmidt-like process. Rotations estimated as 6D representations are more accurate than Euler angles and quaternions (Zhou et al., 2019). As such, the 6D representation has become widely adopted for human pose and shape estimation tasks.

Upon closer examination, we observe that common orthogonalization procedures, including the Gram-Schmidt process and SVD, are problematic for neural network training. Specifically, orthogonalization makes updates with standard first-order gradient descent ambiguous, thereby adding to the learning difficulty and reducing training efficiency. An explicit derivation shows that each gradient can be decomposed into potentially conflicting terms, i.e. in opposing directions, especially at the early stages of training. This finding is related not to numerical instabilities but rather to the internal operations for orthogonalizations like orthogonal projection and cross-product. In addition, incorporating orthogonalizations may introduce extremely large gradients that destabilize training.

To relieve these issues, we advocate removing orthogonalizations from the learning process. Instead, we propose to learn **P**seudo **Ro**tation **M**atrices (PRoM), or unorthogonalized rotation matrices. Orthogonalization is applied post-hoc during inference. A key advantage of PRoM is that the update of each element in the predicted matrix is based only on the difference with respect to its corresponding ground truth; this fully avoids both ambiguity and numerical instability. It also ensures that the prediction of each matrix element remains independent, as opposed to being coupled in the orthogonalization process. By removing the orthogonalization, PRoM converges faster than methods that keep the orthogonalization. Furthermore, PRoM is guaranteed to converge to a better solution, due

Figure 1: Pipeline for rotation learning with neural networks. The grey part denotes the previous learning process which incorporates orthogonalization in either $r$ or $g$ to obtain the orthogonalized matrix $\hat{\mathbf{R}}_1$. However, we recommend using unorthogonalized matrices $\hat{\mathbf{R}}_0$ to compute both losses, which demonstrates faster convergence and better loss results.

to the non-local-injectivity of orthogonalization functions. Simply put, there are multiple estimates corresponding to the same orthogonalized matrix, which becomes problematic for training.

Estimated 3D rotations are often only intermediate outputs that are then applied downstream. In such cases, the learning tends to be end-to-end, with supervision from the downstream task. For example, in 3D human pose estimation, rotations are used for forward kinematics, but the supervision comes not as ground truth rotation matrices but rather as 3D body poses. To ensure valid rotations, existing works (Zhou et al., 2019; Levinson et al., 2020) orthogonalize estimated rotation matrices during both training and inference. In this scenario, we break the convention and recommend passing unorthogonalized matrices to the downstream tasks for end-to-end learning; orthogonalization is applied only as post-processing during inference. The validity of our gradient analysis and optimization holds even when orthogonalization is in the middle or at the end. Empirically, we show that integrating PRoM into body/hand/point cloud pose estimation tasks converges faster for training and leads to better performance for the downstream task. Summarizing our contributions:

- We uncover an ambiguous and explosive gradient issue when incorporating orthogonalization into network training. To mitigate this, we recommend removing orthogonalization from learning and representing the rotation with a *'pseudo'* rotation matrix (PRoM).

- We show, via derivation, why PRoM can converge faster and to a better solution than pipelines with orthogonalization, due to the non-local-injectivity of orthogonalization.

- We empirically demonstrate the superiority of PRoM on several real-world tasks with different combinations of supervision. By changing few lines of code, we achieve state-of-the-art results on several large-scale benchmarks.

## 2    RELATED WORK

**Learning for rotations.** 3D rotations can be described using 3 Euler angles, axis-angles, or quaternions (Rieger & Van Vliet, 2004). However, several works (Grassia, 1998; Rieger & Van Vliet, 2004; Saxena et al., 2009; Knutsson et al., 2011; Zhou et al., 2019) have pointed out that the parameterization of 3D rotations with four or fewer dimensions is discontinuous and non-ideal for learning. To address the discontinuity issue, (Zhou et al., 2019) proposed the continuous 6D representation with a Gram-Schmidit-like process by dropping the last column of the full $3 \times 3$ rotation matrix and recover it with a cross-product operation. In a similar use of matrices and orthogonalizations, (Levinson et al., 2020) recommended using SVD-based orthogonalization as it is a better approximation than Gram-Schmidt process under Gaussian noise. In our paper, we focus on the learning of rotation matrices from the perspective of gradient and optimization. By showing the analysis, we prove that removing the orthogonalizations during learning benefits the convergence of rotation loss.

**3D human body/hand pose and shape estimation**. 3D rotations are critical intermediate representations for downstream tasks such as body/hand pose and shape estimation. The accuracy of predicted 3D rotations largely influences the quality of 3D mesh. The classic HMR (Kanazawa et al., 2018) adopted axis-angle rotation representation, while most subsequent works (Kolotouros et al., 2019; Kocabas et al., 2020; Choi et al., 2021; Li et al., 2022) used the continuous 6D representation. Previous works mainly focus on how to utilize different sources of supervision (2D and 3D keypoint locations) or network design. However, by only changing the representation of rotation with several lines of code from 6D to PRoM, we achieve state-of-the-art results.

## 3 AN ANALYSIS ON ORTHOGONALIZATION

### 3.1 PRELIMINARIES

Consider a representation $\theta$ for an $n$-dimensional rotation which is estimated by a neural network $f_{\boldsymbol{w}}$ with the input $\mathbf{X}$, i.e., $\hat{\theta} = f_{\boldsymbol{w}}(\mathbf{X})$. Without any assumption on the form of $\theta$, the mapping of $\theta$ to an $n \times n$ rotation matrix $\mathbf{R} \in \mathbb{R}^{n \times n}$ can be defined by $r : \theta \to \mathbf{R}$ (see Fig. 1). For example, if $\theta$ is an axis-angle representation, then $r$ is the Rodrigues' rotation formula; if $\theta$ is already an $n \times n$ matrix, then $r$ is simply an identity mapping.

Regressions with neural networks are typically unconstrained, so if the predicted $\hat{\theta}$ is an $n \times n$ matrix, it is unlikely to be a valid rotation matrix and a subsequent orthogonalization is necessary. We denote the unorthogonalized estimate as a "pseudo" rotation matrix and the orthogonalized version as $\mathbf{R}_1$; additionally, let $g : \mathbf{R}_0 \to \mathbf{R}_1$ denote the orthogonalization. Two common orthogonalization methods are based on Singular Value Decomposition (SVD) and the Gram-Schmidt Process.

Given a matrix $\mathbf{P} \in \mathbb{R}^{n \times n}$, **Singular Value Decomposition** decomposes $\mathbf{P}$ into three matrices, i.e. $\mathbf{P} = \mathbf{U}\Sigma\mathbf{V}^{\mathsf{T}}$, where $\mathbf{U} \in \mathbb{R}^{n \times n}$, $\mathbf{V} \in \mathbb{R}^{n \times n}$ are orthogonal matrices and $\Sigma \in \mathbb{R}^{n \times n}$ is a diagonal matrix with all positive values on its diagonals. The orthogonalized version of matrix $\mathbf{P}$ can then be defined as $\mathbf{R}_{\text{SVD}} = g_{\text{SVD}}(\mathbf{P}) = \mathbf{U}\mathbf{V}^{\mathsf{T}}$.

The **Gram-Schmidt** process sequentially projects each column vector to be orthogonal to the previous. Consider a matrix $\mathbf{P} \in \mathbb{R}^{n \times n} = [\boldsymbol{p}_1, \ldots, \boldsymbol{p}_i, \ldots, \boldsymbol{p}_n]$ with column vectors $\boldsymbol{p}_i \in \mathbb{R}^n$. It can be orthogonalized to $\mathbf{R}_{\text{GS}} = g_{\text{GS}}(\mathbf{P}) = [\boldsymbol{q}_1, \ldots, \boldsymbol{q}_i, \ldots, \boldsymbol{q}_n]$ with column vectors $\boldsymbol{q}_i \in \mathbb{R}^n$ as follows:

$$\boldsymbol{q}_i = \begin{cases} N(\boldsymbol{p}_1) & \text{if } i = 1 \\ N(\boldsymbol{p}_i - \sum_{j=1}^{i-1}(\boldsymbol{q}_j \cdot \boldsymbol{p}_i)\boldsymbol{q}_j) & \text{if } 2 \leq i \leq n \end{cases}, \tag{1}$$

where $N(\cdot)$ denotes a vector normalization, i.e. $N(\boldsymbol{p}) = \frac{\boldsymbol{p}}{|\boldsymbol{p}|}$ and $|\boldsymbol{p}|$ is the magnitude of the vector $\boldsymbol{p}$.

### 3.2 6D REPRESENTATION FOR 3D ROTATIONS

Since $\mathbf{R} \in \mathbb{R}^{n \times n}$ is in the set of $SO(n)$, it has only $n$ degrees of freedom and can be expressed more compactly. (Zhou et al., 2019) proposed an alternative representation with $n^2 - n$ parameters by simply dropping the last column vector of the rotation matrix $\mathbf{R}$. As our interest is primarily in 3D rotations, we follow (Zhou et al., 2019) and refer to this representation as a '6D' representation, even though it is general for $n$-dimensional rotations.

For a given rotation in the 6D representation, $\theta_{\text{6D}} = [\boldsymbol{t}_1', \boldsymbol{t}_2']$, where $\boldsymbol{t}_i' \in \mathbb{R}^3$ are column vectors of $\theta_{\text{6D}}$, the full rotation matrix $\mathbf{R}_{\text{6D}}$ can be determined by $\mathbf{R}_{\text{6D}} = r_{\text{GS}}(\theta_{\text{6D}}) = [\boldsymbol{r}_1', \boldsymbol{r}_2', \boldsymbol{r}_3']$. The column vectors $\boldsymbol{r}_i' \in \mathbb{R}^3$ are obtained by a Gram-Schmidt-like process:

$$\boldsymbol{r}_i' = \begin{cases} N(\boldsymbol{t}_1') & \text{if } i = 1 \\ N(\boldsymbol{t}_2' - (\boldsymbol{r}_1' \cdot \boldsymbol{t}_2')\boldsymbol{r}_1') & \text{if } i = 2 \\ \boldsymbol{r}_1' \times \boldsymbol{r}_2' & \text{if } i = 3. \end{cases} \tag{2}$$

The difference between $r_{\text{GS}}$ (Eq. 2) and the standard Gram-Schmidt $g_{\text{GS}}$ (Eq. 1) is that the last column vector $\boldsymbol{r}_3'$ is determined by cross product of $\boldsymbol{r}_1'$ and $\boldsymbol{r}_2'$. One advantage of the 6D representation, as claimed in (Zhou et al., 2019), is Eq. 2 directly guarantees that $\mathbf{R}_{\text{6D}}$ is a special orthogonal matrix, so there is no need for orthogonalization. Currently, the 6D representation is widely used in human body and hand pose and shape estimation tasks (Kolotouros et al., 2019; Li et al., 2022).

### 3.3 DOWNSTREAM TASKS AND LEARNING

In many tasks, the estimated $\hat{\theta}$ is an intermediate output; it is transformed to a rotation matrix $\hat{\mathbf{R}}_1$ for a downstream, task-specific output $\hat{\mathbf{Y}}$. Let $h : (\mathbf{R}_1, \mathbf{X}_{\mathbf{Y}}) \to \hat{\mathbf{Y}}$ denote the downstream computation where $\mathbf{X}_{\mathbf{Y}}$ is the input of the downstream task. For example, in human body mesh recovery, $h$ is the forward kinematics via statistic body models like SMPL (Loper et al., 2015), and $\hat{\mathbf{Y}}$ is the estimated 3D mesh vertices, while $\mathbf{X}$ is the input image and $\mathbf{X}_{\mathbf{Y}}$ is the estimated shape and camera

parameters. Since the downstream task of mesh recovery is widely used throughout this paper, we refer the reader to additional details in Appendix A.1.

The functions $r$, $g$, and $h$ form a computational graph from $\theta$ to $\mathbf{R}_1$ to the target output $\hat{\mathbf{Y}} = h(g(r(\theta)))$ (see Fig. 1). During learning, there may exist two losses: one on the rotation, $\mathcal{L}_\theta$, and one on the downstream tasks, $\mathcal{L}_\mathbf{Y}$. For tasks where the intermediate rotation ground truth is not available, $\mathcal{L}_\mathbf{Y}$ can also be applied as the only form of supervision. For the tasks ending with predicting rotations, only $\mathcal{L}_\theta$ is considered. In the paper, we consider all three conditions for comprehensiveness. In general, the standard practice (Zhou et al., 2019; Levinson et al., 2020) is to calculate element-wise losses $\mathcal{L}_{\text{ele}}$ for both rotation and downstream tasks:

$$\mathcal{L} = \mathcal{L}_\theta + \mathcal{L}_\mathbf{Y} = \mathcal{L}_{\text{ele}}(\mathbf{R}, \hat{\mathbf{R}}_1) + \mathcal{L}_{\text{ele}}(\mathbf{Y}, h(\hat{\mathbf{R}}_1)) \tag{3}$$

where $\mathbf{R}$ and $\mathbf{Y}$ are the corresponding ground truth for the rotation and downstream output, respectively, when available. The loss $\mathcal{L}_{\text{ele}}$ can be an MSE or MAE loss.

## 3.4 GRADIENT ANALYSIS

We consider the case of 3D rotations. The gradients of neural network weights $\boldsymbol{w}$ from the loss $\mathcal{L}$ in Eq. 3 can be formulated as

$$\frac{\partial \mathcal{L}}{\partial \boldsymbol{w}} = \left( \frac{\partial \mathcal{L}_\theta}{\partial g(r(\theta))} + \frac{\partial \mathcal{L}_\mathbf{Y}}{\partial h(g(r(\theta))} \nabla h \right) \nabla g \nabla r \nabla f, \tag{4}$$

where $\frac{\partial \mathcal{L}_\theta}{\partial g(r(\theta))}$ and $\frac{\partial \mathcal{L}_\mathbf{Y}}{\partial h(r(f(x))} \nabla h$ are determined by the type of loss (e.g., MSE loss) and the downstream computation (e.g., forward kinematics in human mesh recovery); $\nabla g$ and $\nabla r$ denote the gradients of the orthogonalization algorithm and rotation representation (e.g., 6D from (Zhou et al., 2019)). Here we focus on analyzing $\nabla g \nabla r$ since this depends on the rotation representation, which is also multiplied with $\nabla h$ in the downstream task.

In an orthogonalized matrix, the columns have to be orthonormal and methods such as Gram-Schmidt and SVD-based orthogonalization mainly consist of column-wise operations except for vector normalization. Therefore, we analyze the gradients on a column basis and study the gradients of the orthonormal columns $g(r(f_{\boldsymbol{w}}(\mathbf{X}))) = \{\boldsymbol{r}'_1, \boldsymbol{r}'_2, \boldsymbol{r}'_3\}$ with respect to the unorthogonalized columns or the direct output of the network $f_{\boldsymbol{w}}(\mathbf{X}) = \{\boldsymbol{t}'_1, \boldsymbol{t}'_2, \boldsymbol{t}'_3\}$ where $\boldsymbol{r}'_i$ and $\boldsymbol{t}'_i$ is the $i^{\text{th}}$ column in the corresponding matrix. Consider the gradient of the rotation loss $\mathcal{L}_\theta$ w.r.t. the first unorthogonalized column $\boldsymbol{t}'_1$, which can be expressed as

$$\frac{\partial \mathcal{L}_\theta}{\partial \boldsymbol{t}'_1} = (\boldsymbol{r}'_1 - \boldsymbol{r}_1)^\intercal \frac{\partial \boldsymbol{r}'_1}{\partial \boldsymbol{t}'_1} + (\boldsymbol{r}'_2 - \boldsymbol{r}_2)^\intercal \frac{\partial \boldsymbol{r}'_2}{\partial \boldsymbol{t}'_1} + (\boldsymbol{r}'_3 - \boldsymbol{r}_3)^\intercal \frac{\partial \boldsymbol{r}'_3}{\partial \boldsymbol{t}'_1} \tag{5}$$

In the equation above, $\{\boldsymbol{r}_1, \boldsymbol{r}_2, \boldsymbol{r}_3\}$ are the corresponding ground truth vectors. For the 6D representation, $f_w(\mathbf{X}) = \{\boldsymbol{t}'_1, \boldsymbol{t}'_2\}$ as it generates $\boldsymbol{r}'_3$ by cross product. However, it does not influence our analysis. We provide explicit derivations of Eq. 5 for 6D (Gram-Schmidt-like)-based orthogonalization in Appendix A.2.

### 3.4.1 UPDATE AMBIGUITY

We claim that the shortest path to update $\boldsymbol{r}'_1$ to $\boldsymbol{r}_1$ should be consistent with $(\boldsymbol{r}'_1 - \boldsymbol{r}_1)^\intercal \frac{\partial \boldsymbol{r}'_1}{\partial \boldsymbol{t}'_1}$. Yet SGD-based optimization updates according to Eq. 5, where the gradient for $\boldsymbol{t}'_1$ is a weighted sum of the differences between *all* the orthonormal columns and their respective ground truths, i.e. with additional terms $(\boldsymbol{r}'_2 - \boldsymbol{r}_2)^\intercal \frac{\partial \boldsymbol{r}'_2}{\partial \boldsymbol{t}'_2}$ and $(\boldsymbol{r}'_3 - \boldsymbol{r}_3)^\intercal \frac{\partial \boldsymbol{r}'_3}{\partial \boldsymbol{t}'_3}$. Moreover, the extra terms are non-negligible and may give opposite directions compared to $(\boldsymbol{r}'_1 - \boldsymbol{r}_1)^\intercal \frac{\partial \boldsymbol{r}'_1}{\partial \boldsymbol{t}'_1}$. In short, this is an *ambiguous update* and results in lower training efficiency, especially at the beginning of training. More details on the ambiguous update are provided in Appendix A.3 and empirical verification are in Sec. 5.1.

One could contend that the training speed is inconsequential as long as it ultimately yields optimal outcomes. Sec. 4.2 proves why including orthogonalization may also lead to suboptimal results.

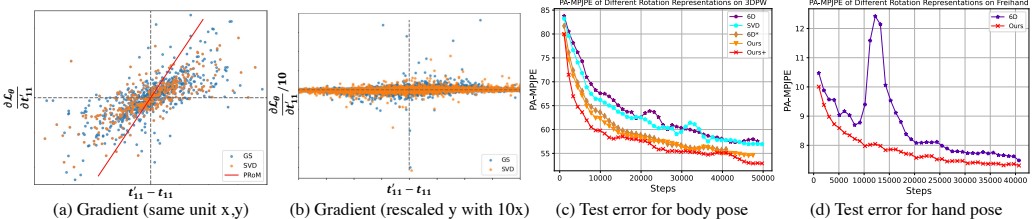

(a) Gradient (same unit x,y)    (b) Gradient (rescaled y with 10x)    (c) Test error for body pose    (d) Test error for hand pose

Figure 2: (a) Gradient value w.r.t. $(t'_{11} - t_{11})$. PRoM shows consistent gradient for any given x, whereas 6D- and SVD-based methods show the ambiguous update with diverse values for the same x. (b) Gradient value w.r.t. $(t'_{11} - t_{11})$ with rescaled y to 10 times larger. We can observe extremely large value that indicate the existence of gradient explosion. (c) Test error on 3DPW with different rotation representations. $6D^*$ means removing orthogonalizations during the learning of 6D representation. Ours+ indicates using a larger learning rate. (d) Test error on FreiHAND with different rotation representations. The instability of 6D gradients causes a jitter during training.

### 3.4.2 GRADIENT EXPLOSION

Orthogonalization operations like Gram-Schmidt and SVD are also prone to gradient explosion. For example, in Gram-Schmidt-based orthogonalizations, $\frac{\partial r'_2}{\partial t'_1}$ may become extremely large. From Eqs. 1 and 2, if we denote $r''_2 = t'_2 - (r'_1 \cdot t'_2)r'_1$, then $\frac{\partial r'_2}{\partial t'_1}$ can be written as

$$\frac{\partial r'_2}{\partial t'_1} = \frac{\partial r'_2}{\partial r''_2}\frac{\partial r''_2}{\partial t'_1} = \nabla N(r''_2)\frac{\partial r''_2}{\partial t'_1} = -\frac{1}{|r''_2|}(\mathbf{I} - \frac{r''_2(r''_2)^\intercal}{|r''_2|^2})\frac{\partial r''_2}{\partial t'_1},\tag{6}$$

where $\nabla N(\cdot)$ is the gradient of normalization function. When $t'_1$ and $t'_2$ are parallel, $|r''_2|$ approaches 0, leading to very large gradients that destabilize backpropagation. More detailed derivations are shown in Appendix A.2.

With SVD, the gradients depend on a matrix $\mathbf{K}$ with elements $\mathbf{K}_{ij} = \frac{1}{\lambda_i - \lambda_j}$ when $i \neq j$ and 0 otherwise, where $\lambda_i$ and $\lambda_j$ denote the $i^{\text{th}}$ and $j^{\text{th}}$ eigenvalues of the matrix being orthogonalized (Ionescu et al., 2015). This leads to either zero components, or, when $\lambda_i$ is close to $\lambda_j$, very large components that result in gradient explosion (Wang et al., 2021).

Empirically, we observe large gradients in the synthetic and real-world experiments when incorporating SVD or Gram-Schmidt for orthogonalization (see details in Fig. 2 (b)). From a practical perspective, the instability can be managed with a small learning rate and a larger batch size, but this slows down the learning process.

## 4 METHOD

### 4.1 PSEUDO ROTATION MATRIX (PROM)

Given that orthogonalization may cause update ambiguities and destabilize training, we advocate representing rotations with a *'pseudo'*-rotation matrix (PRoM) $\mathbf{R}_0$. In PRoM, we simply remove the orthogonalization function $g$ and apply the estimated orthogonalized matrix $\hat{\mathbf{R}}_0$ in place of the orthogonalized $\hat{\mathbf{R}}_1$. In this case, both $r(\cdot)$ and $g(\cdot)$ become identify mappings, so the loss from Eq. 3 simplifies to the following:

$$\mathcal{L}' = \mathcal{L}'_\theta + \mathcal{L}'_\mathbf{Y} = \mathcal{L}_{\text{ele}}(\mathbf{R}, \hat{\mathbf{R}}_0) + \mathcal{L}_{\text{ele}}(\mathbf{Y}, h(\hat{\mathbf{R}}_0)).\tag{7}$$

Subsequently, the gradients in Eq. 4 become

$$\frac{\partial \mathcal{L}'}{\partial \boldsymbol{w}} = \left(\frac{\partial \mathcal{L}'_\theta}{\partial \theta} + \frac{\partial \mathcal{L}'_\mathbf{Y}}{\partial h(\theta)}\nabla h\right)\mathbf{I}\,\nabla f.\tag{8}$$

From Eq. 4, $\nabla g\nabla r$ simplifies into an identity matrix $\mathbf{I}$ and relieves the issue of the ambiguity and instability of original gradients. With more consistent and stable gradients, PRoM converges faster than when incorporating SVD or Gram-Schmidt; it can also tolerate higher learning rates up to 5 times.

## 4.2 Optimization Analysis

In addition to faster convergence, PRoM converges to a better solution than when incorporating orthogonalization. In the following analysis, we abuse notation and use $\hat{\mathbf{R}}_0, \hat{\mathbf{R}}_1$ to represent the unorthogonalized and orthogonalized matrices respectively of $m$ samples instead of one sample, i.e., $\hat{\mathbf{R}}_1 \in \mathbb{R}^{m \times n^2}$ and $\hat{\mathbf{R}}_0 \in \mathbb{R}^{m \times n^2}$. In addition, $\mathcal{R}^* \subset \mathbb{R}^{m \times n^2}$ is the set of all optimal matrices consisting of $m$ ground truth matrices. We consider an arbitrary total loss criterion $L$ that takes the output matrix $\hat{\mathbf{R}}_1$ for any orthogonalization-incorporated methods and $\hat{\mathbf{R}}_0$ for the proposed PRoM.

**Theorem 1.** *For any optimal matrix* $\mathbf{R}^* \in \mathcal{R}^*$ *and* $i \in \{0, 1\}$,

$$L(\hat{\mathbf{R}}_i) \leq L(\mathbf{R}^*) + \epsilon C \psi(\mathbf{B}_i), \quad \text{where} \quad C = E\psi(\mathbf{D}). \tag{9}$$

Theorem 1 states that the loss from any estimated rotation matrix, $L(\hat{\mathbf{R}}_i)$ converges at a rate that depends on the term $\psi(\mathbf{B}_i)$ (see Theorem 2 below). More specifically, in Theorem 1, define $\epsilon = \sup_{i \in \{0,1\}} \|(\frac{\partial L(\hat{\mathbf{R}}_i)}{\partial \boldsymbol{w}})^\intercal\|$, where $\|\cdot\|$ represents the Euclidean norm. $\epsilon \to 0$ as the number of iterations increases in many gradient-based optimizers under mild conditions (Bertsekas, 1997) including stochastic gradient descent (Fehrman et al., 2020; Lei et al., 2019; Mertikopoulos et al., 2020). $E$ is a constant defined as $E = \inf_{\mathbf{R}^* \in \mathcal{R}^*} \sup_{\hat{\mathbf{R}} \in \mathcal{R}} \|\mathbf{R} - \hat{\mathbf{R}}\|$, where $\mathcal{R}$ is the set of all $\hat{\mathbf{R}}_i$ encountered during training. The term $\psi(\mathbf{M})$, for any real matrix $\mathbf{M}$, define $\psi(\mathbf{M}) = 1/\sqrt{\lambda_{\min}(\mathbf{MM}^\intercal)}$ if $\lambda_{\min}(\mathbf{MM}^\intercal) \neq 0$ and $\psi(\mathbf{M}) = \infty$ if $\lambda_{\min}(\mathbf{MM}^\intercal) = 0$.

In Theorem 1, matrices $\mathbf{D}$ and $\mathbf{B}_i$ are gradient matrices:

$$\mathbf{D} = \frac{\partial \hat{\mathbf{R}}_0}{\partial \boldsymbol{w}} \in \mathbb{R}^{mn^2 \times d} \qquad \text{and} \qquad \mathbf{B}_i = \frac{\partial \hat{\mathbf{R}}_i}{\partial \hat{\mathbf{R}}_0} \in \mathbb{R}^{mn^2 \times mn^2}, \quad i \in \{0, 1\}, \tag{10}$$

where $d$ is the number of parameters as $\boldsymbol{w} \in \mathbb{R}^d$. Theorem 1 is established under the assumption that the training loss, e.g., $\boldsymbol{w} \to L(\hat{\mathbf{R}}_i)$, is *non-convex* while the loss criterion $\hat{\mathbf{R}}_i \to L(\hat{\mathbf{R}}_i)$, is differentiable and convex. These assumptions are satisfied by using neural networks with common loss criteria such as the MSE loss. The detailed proof is given in Appendix A.4.

We can ensure that $\lambda_{\min}(\mathbf{DD}^\intercal) \neq 0$ by increasing the width of the neural network (Kawaguchi & Bengio, 2019; Kawaguchi et al., 2022). As such, Theorem 1 shows that $L(\hat{\mathbf{R}}_i)$ can converge to the optimal loss $L(\mathbf{R}^*)$ at a rate specified by $\psi(\mathbf{B}_i)$. The property of $\psi(\mathbf{B}_i)$ depends on $i$, i.e., if the matrix is orthogonalized or not, and is given in Theorem 2.

**Theorem 2.** *For any* $\hat{\mathbf{R}}_0$, $\psi(\mathbf{B}_0) = 1$ *and* $\psi(\mathbf{B}_1) = \infty$ *for any* $g$ *that is not locally injective at* $\hat{\mathbf{R}}_0$.

We give proof of Theorem 2 in Appendix A.5. This theorem states for any orthogonalization incorporated method, $\psi(\mathbf{B}_1)$ is infinity, which makes the term $\epsilon C \psi(\mathbf{B}_1)$ non-negligible. However, by removing the orthogonalization during the training, $\epsilon C \psi(\mathbf{B}_0)$ approaches 0.

Theorems 1 and 2 together establish that gradient descent with PRoM is faster than approaches that incorporate orthogonalization. Furthermore, using PRoM can make the loss converge to the optimal value, whereas the orthogonalization incorporated methods with $i = 1$ may fail to yield optimal outcomes since $\psi(\mathbf{B}_1) = \infty$. The difference between the two arises from the non-local-injectivity of $g$, the orthogonalization procedure (see A.6 for a simple illustration). By removing the orthogonalization from the computational graph, as we have done with PRoM, the gradient descent is proven to find the optimal solution.

For the downstream task, we can easily obtain a similar conclusion that $L(\hat{\mathbf{Y}}_0)$ will converge to the optimal results while $L(\hat{\mathbf{Y}}_1)$ may fail to do so. We give proof in Appendix A.7.

## 5 Experiments

### 5.1 Empirical Verification

**Update Ambiguity and Gradient Explosion.** We compare the gradients from the Gram-Schmidt and SVD orthogonalizations with direct MSE loss gradients on pseudo rotation matrices. For better

| Method | Human3.6M | | 3DPW | | | AGORA | | FreiHAND | |
|---|---|---|---|---|---|---|---|---|---|
| | MPJPE ↓ | PA-MPJPE ↓ | MPJPE ↓ | PA-MPJPE ↓ | MPVPE ↓ | MPJPE ↓ | MVE ↓ | PA-MPVPE ↓ | PA-MPJPE ↓ |
| MANO CNN | - | - | - | - | - | - | - | 10.9 | 11.0 |
| SPIN | - | 41.1 | 96.9 | 59.2 | 116.4 | 153.4 | 148.9 | - | - |
| VIBE | 65.9 | 41.5 | 93.5 | 56.5 | 113.4 | - | - | - | - |
| HybrIK | 54.4 | 34.5 | 80.0 | 48.8 | 94.5 | 77.0 | 73.9 | - | - |
| CLIFF | 47.1 | 32.7 | 69.0 | 43.0 | 81.2 | 81.0 | 76.0 | 6.6 | 6.8 |
| CLIFF + PRoM | **43.8** (-3.3) | **30.4** (-2.3) | **67.6** (-1.4) | **42.0** (-1.0) | **79.2** (-2.0) | **65.0** | **61.0** | **6.4** (-0.2) | **6.5** (-0.3) |

Table 1: Evaluation of state-of-the-art methods on Human3.6M (Ionescu et al., 2013), 3DPW (Von Marcard et al., 2018), AGORA (Patel et al., 2021) and FreiHAND (Zimmermann et al., 2019). We achieve the best results among all the methods on 3D human body and hand pose benchmarks.

| | rotation recovery | | | point cloud pose estimation | | |
|---|---|---|---|---|---|---|
| | Mean(°) | Max(°) | Std(°) | Mean(°) | Max(°) | Std(°) |
| Axis-Angle | 3.69 | 179.22 | 5.99 | 11.93 | 179.7 | 21.35 |
| Euler | 6.98 | 179.95 | 17.31 | 14.13 | 179.67 | 23.8 |
| Quat | 3.32 | 179.93 | 5.97 | 9.03 | 179.66 | 16.33 |
| 6D | 0.49 | 1.98 | 0.27 | 2.85 | 179.83 | 9.16 |
| PRoM | **0.37** | **1.86** | **0.22** | **2.13** | 179.53 | **7.87** |

(a)

| | MPJPE ↓ | PA-MPJPE ↓ | MPVPE ↓ |
|---|---|---|---|
| SMPLify | - | 139.5 | - |
| SPIN | - | 52.0 | - |
| EFT | - | 49.3 | - |
| CLIFF* | 52.8 | 32.8 | 61.5 |
| CLIFF* + PRoM | **49.5** (-3.2) | **29.9** (-2.9) | **56.9** (-4.6) |

(b)

Table 2: (a) Comparison of methods through the mean, maximum, standard deviation of errors of rotation recovery (left) and point cloud pose estimation test (right). Compared with traditional methods and 6D representation, our method has the smallest errors. (b) Evaluation of optimization-based methods on 3DPW providing the 2D ground truth. CLIFF* denotes the CLIFF annotator that refines 3D rotation by 2D ground truth. Our method achieves the best result with a 7% reduction in PA-MPJPE.

visualization, we pick the first element of $t'_1$, i.e., $t'_{11}$, and draw the gradient value w.r.t. the difference between $t'_{11}$ and its ground truth $t_{11}$. We randomly generate a ground truth rotation matrix $\mathbf{R}$, based on which we construct the predicted rotation matrix $\hat{\mathbf{R}}$ with $t'_{ij} = r_{ij} + \mathcal{N}(0, \sigma^2)$ following (Levinson et al., 2020). We run 10K iterations for three different $\sigma$s to show the distribution of the gradients on $t'_{11}$. The full results are in Appendix A.8.

The result with $\sigma = 0.5$ are displayed in Fig. 2 (a) and (b), where the x- and y-axis of each subfigure are $(t'_{11} - t_{11})$ and its gradient $\frac{\partial \mathcal{L}_\theta}{\partial t'_{11}}$ for one run. In Fig. 2 (a), the x- and y-axis share the same unit whereas the unit of y-axis in Fig. 2 (b) is 10 times larger than x-axis. From Fig. 2 (a), we can see that incorporating Gram-Schmidt or SVD will give diverse gradients, some of which may be opposite the direction of $(t'_{11} - t_{11})$ (e.g., datapoints in the second and fourth quadrants) whereas PRoM will provide consistent gradients. From Fig. 2 (b), we can observe many "flying" datapoints that demonstrate extremely large values, which indicates the gradient explosion issue.

**Minimum Eigenvalues of $\mathbf{B}_i \mathbf{B}_i^\mathsf{T}$.** It's clear that for the proposed PRoM, $\lambda_{\min}(\mathbf{B}_0 \mathbf{B}_0^\mathsf{T}) = 1$. To empirically verify $\lambda_{\min}(\mathbf{B}_1 \mathbf{B}_1^\mathsf{T}) \to 0$, we record the matrices during the training of pose and shape estimation tasks and calculate $\lambda_{\min}(\mathbf{B}_1 \mathbf{B}_1^\mathsf{T})$. For both 6D-based and SVD-based learning, $\lambda_{\min}(\mathbf{B}_1 \mathbf{B}_1^\mathsf{T})$ is smaller than 1e-18, which verifies that $\lambda_{\min}(\mathbf{B}_1 \mathbf{B}_1^\mathsf{T}) \to 0$.

## 5.2 COMPARISON WITH THE STATE-OF-THE-ART

To verify the effectiveness of the proposed rotation representation PRoM, we conduct extensive experiments on large-scale benchmarks with different tasks. Considering that different tasks have different supervision, we split the tasks into three categories: tasks supervised by both ground truth rotations and downstream outputs (Sec. 5.2.1), tasks ended with predicting rotations (Sec. 5.2.2), tasks with ground truth downstream labels but without intermediate rotation labels (Sec. 5.2.3). For the orthogonalization as post-processing during inference, we use SVD-based orthogonalization unless specially mentioned. We provide detailed experimental settings for all experiments in Appendix A.9.

| | $\mathcal{L}_\theta$ | $\mathcal{L}_\mathbf{Y}$ | post | PA-MPJPE↓ |
|---|---|---|---|---|
| Zhou et al. (2019) | $r_\mathrm{GS}$ | $r_\mathrm{GS}$ | ✗ | 57.5 |
| Levinson et al. (2020) | $g_\mathrm{SVD}$ | $g_\mathrm{SVD}$ | ✗ | 56.7 |
| | id. | $r_\mathrm{GS}$ | $r_\mathrm{GS}$ | 57.1 |
| | $r_\mathrm{GS}$ | id. | $r_\mathrm{GS}$ | 55.8 |
| | id. | id. | $r_\mathrm{GS}$ | 55.6 |
| Ours | id. | id. | $g_\mathrm{SVD}$ | **54.8** |

| LR | (Zhou et al., 2019) | (Levinson et al., 2020) | PRoM |
|---|---|---|---|
| 1e-4 | 58.3 | 56.7 | 54.8 |
| 2e-4 | - | - | 53.4 |
| 5e-4 | - | - | 52.6 |
| 8e-4 | - | - | - |

(a)                                                             (b)

Table 3: (a) Ablation study of removing orthogonalizations in only $\mathcal{L}_\theta$ or $\mathcal{L}_\mathbf{Y}$. 'id.' means identity mapping. We see that using identity mapping in $\mathcal{L}_\mathbf{Y}$ is more critical. (b) Different methods with varying learning rates. '-' denotes NaN. We show that PRoM can tolerate up to 5 times larger learning rates and gives better performance.

### 5.2.1 BOTH $\mathcal{L}_\theta$ AND $\mathcal{L}_\mathbf{Y}$

3D human body and hand pose and shape estimation tasks are challenging and involve both rotation and downstream outputs. Previous works mainly focus on network designs and additional information help, however, we show that by simply changing the rotation representation to the proposed PRoM, we obtain significant improvement and achieve state-of-the-art results.

For both 3D body and hand pose estimation, we follow the network design and training settings of the state-of-the-art method CLIFF (Li et al., 2022). The implementation details can be seen in Appendix A.10. For evaluation, we consider 3D Euclidean distances in millimeters (mm) between predictions and the ground truth: **MPJPE** (Mean Per Joint Position Error), **PA-MPJPE** (Procrustes-Aligned MPJPE), and **PVE** (Per Vertex Error) on the human mesh surfaces. For all these three metrics, lower values indicate better performance.

**3D Body Pose and Shape Estimation** Following previous work (Kolotouros et al., 2019; Li et al., 2022; Lin et al., 2021), we train the network with a mixture of datasets, including Human3.6M (Ionescu et al., 2013), MPI-INF-3DHP (Mehta et al., 2017), 3DPW (Von Marcard et al., 2018), MSCOCO (Lin et al., 2014), and MPII (Andriluka et al., 2014), using the pseudo ground truth provided by the CLIFF annotator (Li et al., 2022) for 2D datasets. Evaluation is performed on the indoor dataset Human3.6M, the outdoor dataset 3DPW, and the synthetic dataset AGORA.

We compare our results with state-of-the-art model-based methods(Kolotouros et al., 2019; Kocabas et al., 2020; Li et al., 2021; 2022) on the body pose and shape estimation tasks. From Table 1, we achieve the lowest error for body pose and body mesh. Compared with the existing SOTA method CLIFF (Li et al., 2022), which is also our baseline, we reduce the error by 1.0 - 2.0mm for 3DPW. On Human3.6M, we achieve an impressive 2.3-3.3mm or $7\%$ reduction in error over CLIFF. We also rank 1st on the AGORA (Patel et al., 2021) leaderboard, which sufficiently proves the capability of our method with a powerful backbone on the complex dataset.

**Discussion** As the most challenging task, we discuss how PRoM outperforms baseline 6D-based methods on the body pose and shape estimation task. We do a comparison on a per-sample basis and show that 72% of the samples have lower errors. Since PRoM improves the learning process, it makes sense that it achieves general improvement but not significant improvement on specific samples. We also give visual examples in Appendix A.11.

**3D Hand Pose and Shape Estimation** We perform mixed-dataset training on FreiHAND (Zimmermann et al., 2019), Obman (Hasson et al., 2019), and Interhand2.6M (Moon et al., 2020), and evaluate on FreiHAND. Table 1 shows that we achieve the lowest error among the methods.

**Point Cloud Pose Estimation.** We also verify the effectiveness of PRoM by experimenting on point cloud pose estimation task introduced in (Zhou et al., 2019), which uses 2,290 airplane point clouds to train from ShapeNet (Chang et al., 2015). The test set consists of 400 held-out point clouds augmented with 100 random rotations. We report the results in the right panel of Table. 2 (a). We demonstrate that our method can perform better than formerly popular methods.

### 5.2.2 $\mathcal{L}_\theta$ ONLY

**Recovering Rotations with a Neural Network.** We follow (Zhou et al., 2019) to conduct a rotation recovery based on an auto-encoder architecture to test the quality of recovering rotations. The task ended with predicting the rotations. The input is rotation matrices by uniformly sampling axes and angles. The test set consists of 100,000 randomly generated rotation matrices. The output is evaluated by the geodesic distance between the input matrix and itself. Results in the left panel of Table 2 (a) show that our method achieves the best performance.

### 5.2.3 $\mathcal{L}_\mathbf{Y}$ ONLY

**Pose and Shape Estimation with 2D Ground Truth.** Since 3D rotation and pose annotations are hard to obtain, 2D ground truth locations are usually applied to refine the predictions, especially on in-the-wild datasets like 3DPW. In this case, the task only has downstream supervision. Specifically, we compare our method with optimization-based methods where the mesh vertices can be optimized using 2D ground truth keypoints and rotation matrices are intermediate outputs. In Table 2 (b), our method is 3.3mm better than the baseline method on PA-MPJPE. We are the first to reduce the error of PA-MPJPE to under 30mm on the 3DPW test set with only a change of rotation representation.

### 5.3 ABLATION STUDY

We perform ablations on 3D human body pose and shape estimation and report PA-MPJPE on 3DPW. The settings are all as follows if not stated otherwise. We train a CLIFF-Res50 model on MSCOCO with CLIFF pseudo-GT (Li et al., 2022), as it is fast to train and provides comparable performance on 3DPW. For each ablation, we train with 250 epochs.

**Impact of removing orthogonalizations.** The core of our method is to remove any orthogonalization procedures in the training, including the computation of both $\mathcal{L}_\theta$ and $\mathcal{L}_\mathbf{Y}$. Previous works (Zhou et al., 2019; Levinson et al., 2020) employ $r_{\text{GS}}$ and $g_{\text{SVD}}$ as orthogonalizations in both losses. We explore the impact of removing both or either of them as shown in Table 3 (b). Comparing the results, we can conclude that removing orthogonalization in the computation of either loss improves the results and it's more critical to learn unorthogonalized matrices for the downstream task, which has never been considered before.

**Training speed.** We show the evaluations of PA-MPJPE at every 1k steps of the different methods with a fixed learning rate on the 3DPW test set in Fig. 2 (c). We show that our method consistently trains faster than methods that incorporate orthogonalizations. We also show an interesting training curve when training the hand models in Fig. 2 (d), where the 6D method has a very large jitter while our method trains smoothly. We attribute this jitter to the instable gradients of the 6D representation.

**Learning rate.** We demonstrate the performance of 6D (Zhou et al., 2019), SVD (Levinson et al., 2020), and our method with different learning rate settings to test the gradient stability. The learning rate starts at 1e-4, which is common setting, and increases to 2e-4, 5e-4 and 8e-4. From Table 3 (b), we can see that higher learning rates than 1e-4 result in NaN when incorporating orthogonalizations. However, when increasing the learning rate, our method only explodes at an extremely high learning rate of 8e-4 and achieves significantly better results.

**Different Models.** We show the generalization ability of our method on different human pose models, which adopt different network designs and employ 6D representation. The results are in Appendix A.12 and show that PRoM performs better than 6D representation in all metrics.

## 6 CONCLUSION

We studied the gradients when incorporating orthogonalizations in the learning of rotation matrices and uncovered an ambiguous and explosive gradient issue. We therefore advocate removing orthogonalization procedures from the learning process and instead using pseudo rotation matrices (PRoM). Theoretically, we prove that PRoM will converge faster and to a better solution. By changing lines of code, we demonstrate state-of-the-art results on several benchmarks.

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

## A   APPENDIX

### A.1   DETAILS OF DOWNSTREAM TASKS

Here, we provide the details of downstream tasks with the example of mesh recovery tasks, which are also known as pose and shape estimation tasks. Following (Kanazawa et al., 2018; Kolotouros et al., 2019), the 3D mesh of the body is generated via the Skinned Multi-Person Linear (SMPL) model (Loper et al., 2015), which represents the 3D mesh by shape parameters $\beta \in \mathbb{R}^{10}$ and pose parameters $\theta \in \mathbb{R}^{3K}$ where $K$ is the number of joints. The shape parameters are coefficients of a PCA shape space. Here we focus on the pose parameters, which consist of $K$ rotations. Taking body mesh recovery as an example, the network outputs the relative 3D rotation of $K = 23$ joints. HMR (Kanazawa et al., 2018) utilized axis-angle representations and therefore outputs $3 \times 23$ pose parameters in total. However, the subsequent works (Kolotouros et al., 2019; Li et al., 2021; 2022; Moon & Lee, 2020) all applied 6D representations due to the continuity.

SMPL is an end-to-end and differentiable function that generates a triangulated mesh $\mathbf{Y}_M$ with 6980 vertices by transforming the predefined rest template $\tilde{\mathcal{T}}$ conditioned on pose and shape parameters. We denote this transformation process as $\mathcal{M}$, i.e., $\mathbf{Y}_M = \mathcal{M}(\tilde{\mathcal{T}}, \theta, \beta)$. The 3D keypoints $\mathbf{Y}_J$ are obtained by applying a linear regression on the mesh vertices. We denote this linear transformation as $\mathcal{J}$, i.e., $\mathbf{Y}_J = \mathcal{J}(\mathbf{Y}_M)$. The core operation during $\mathcal{M}$ is deforming the mesh by the given rotations $\mathbf{R}$ with $\mathbf{R} = f(\theta)$, which is essentially matrix multiplication. In this case, the downstream outputs can be concluded as

$$\hat{\mathbf{Y}}_M = \mathcal{M}(\tilde{\mathcal{T}}, f(\hat{\theta}), \hat{\beta}), \ \ \mathbf{Y}_J = \mathcal{J}(\mathcal{M}(\tilde{\mathcal{T}}, f(\hat{\theta}), \hat{\beta})). \tag{11}$$

To ensure the feasibility of the mesh, the previous method applied Rodrigues' rotation formula or a Gram-Schmidt-like process as $f$ on axis-angle and 6D representations, respectively. However, $f(\hat{\theta})$ itself does *not* necessarily need to be orthogonal matrices, but only needs to be $K \times 3 \times 3$ matrices. Through our study, we show that during the learning process, the incorporation of orthogonalizations has a negative influence on the convergence rate and generalization ability on the test set. Also, empirically, we show that removing the orthogonalizations during the training process greatly outperforms keeping them on several large-scale datasets. Note that during inference, we still use orthogonalizations to ensure the feasibility of the mesh since backward propagation is no longer utilized.

For the hand pose and shape estimation task, the only difference is that it utilizes a different parametric model, MANO Romero et al. (2017) with different numbers of pose and shape parameters, but it shares the same pipeline as body pose and shape estimation.

### A.2   FULL GRADIENTS OF 6D-BASED ORTHOGONALIZATION

Recall that $r$ for the 6D representation is given by the Gram-Schmidt-like process in Eq. 2, and $g$ is an identity mapping. Let $\{\boldsymbol{t}'_1, \boldsymbol{t}'_2\}$ be the column vectors in $\theta_{6D}$, $\{\boldsymbol{r}'_1, \boldsymbol{r}'_2, \boldsymbol{r}'_3\}$ be the resulting vectors after applying $f_{GS}$, and $\{\boldsymbol{r}_1, \boldsymbol{r}_2, \boldsymbol{r}_3\}$ be the corresponding ground truth. We denote $\boldsymbol{r}''_2$ as $(\boldsymbol{t}'_2 - (\boldsymbol{r}'_1 \cdot \boldsymbol{t}'_2)\boldsymbol{r}'_1)$, which is the unnormalized value of $\boldsymbol{r}'_2$. The gradient of $\boldsymbol{t}'_1$ from $\mathcal{L}_{6D}$ can be given as

$$\frac{\partial \mathcal{L}_{6D}}{\partial \boldsymbol{t}'_1} = (\boldsymbol{r}'_1 - \boldsymbol{r}_1)^{\mathsf{T}} \frac{\partial \boldsymbol{r}'_1}{\partial \boldsymbol{t}'_1} + (\boldsymbol{r}'_2 - \boldsymbol{r}_2)^{\mathsf{T}} \frac{\partial \boldsymbol{r}'_2}{\partial \boldsymbol{t}'_1} + (\boldsymbol{r}'_3 - \boldsymbol{r}_3)^{\mathsf{T}} \frac{\partial \boldsymbol{r}'_3}{\partial \boldsymbol{t}'_1}, \tag{12}$$

$$\text{where} \ \ \frac{\partial \boldsymbol{r}'_1}{\partial \boldsymbol{t}'_1} = \nabla N(\boldsymbol{t}'_1) = \frac{1}{|\boldsymbol{t}'_1|}(\mathbf{I}_3 - \frac{\boldsymbol{t}'_1(\boldsymbol{t}'_1)^{\mathsf{T}}}{|\boldsymbol{t}'_1|^2}), \tag{13}$$

$$\frac{\partial \boldsymbol{r}'_2}{\partial \boldsymbol{t}'_1} = \frac{\partial \boldsymbol{r}'_2}{\partial \boldsymbol{r}''_2} \frac{\partial \boldsymbol{r}''_2}{\partial \boldsymbol{r}'_1} \frac{\partial \boldsymbol{r}'_1}{\partial \boldsymbol{t}'_1} = \nabla N(\boldsymbol{r}''_2) \frac{\partial \boldsymbol{r}''_2}{\partial \boldsymbol{r}'_1} \frac{\partial \boldsymbol{r}'_1}{\partial \boldsymbol{t}'_1}, \tag{14}$$

$$= -\frac{1}{|\boldsymbol{r}''_2|}(\mathbf{I}_3 - \frac{\boldsymbol{r}''_2(\boldsymbol{r}''_2)^{\mathsf{T}}}{|\boldsymbol{r}''_2|^2})((\boldsymbol{r}'_1 \cdot \boldsymbol{t}'_2)\mathbf{I} + \boldsymbol{r}'_1(\boldsymbol{t}'_2)^{\mathsf{T}}) \frac{\partial \boldsymbol{r}'_1}{\partial \boldsymbol{t}'_1}, \tag{15}$$

$$\frac{\partial \boldsymbol{r}'_3}{\partial \boldsymbol{t}'_1} = \frac{\partial \boldsymbol{r}'_3}{\partial \boldsymbol{r}'_2} \frac{\partial \boldsymbol{r}'_2}{\partial \boldsymbol{t}'_1} + \frac{\partial \boldsymbol{r}'_3}{\partial \boldsymbol{r}'_1} \frac{\partial \boldsymbol{r}'_1}{\partial \boldsymbol{t}'_1} = [\boldsymbol{r}'_1]_{\times} \frac{\partial \boldsymbol{r}'_2}{\partial \boldsymbol{t}'_1} - [\boldsymbol{r}'_2]_{\times} \frac{\partial \boldsymbol{r}'_1}{\partial \boldsymbol{t}'_1}, \tag{16}$$

$\nabla N(\cdot)$ is the gradient from the vector normalization, $\mathbf{I}$ is the identity matrix, and $[\boldsymbol{r}]_{\times}$ is the skew-symmetric matrix of vector $\boldsymbol{r}$.

## A.3 EXPLANATIONS OF UPDATE AMBIGUITY

This claim is established based on two conditions that $(\boldsymbol{r}_i' - \boldsymbol{r}_i)^{\intercal}\frac{\partial \boldsymbol{r}_i'}{\partial \boldsymbol{t}_1'}$ should be isotropic and non-negligible where $i = \{2, 3\}$. For the isotropy, it is guaranteed by the fact that at the initial stage of training, $\boldsymbol{r}_i'$ tends to be randomly generated and therefore $\boldsymbol{r}_i$ can be viewed as isotropic around $\boldsymbol{r}_i'$ together with the fixed $\frac{\partial \boldsymbol{r}_i'}{\partial \boldsymbol{t}_1'}$ given the neural network $f_{\boldsymbol{w}}$ and input $\mathbf{X}$. For the non-negligibility, as the orthogonalizations only incorporate multiplications and additions, this obviously holds. We also provide explicit derivations for $(\boldsymbol{r}_i' - \boldsymbol{r}_i)^{\intercal}\frac{\partial \boldsymbol{r}_i'}{\partial \boldsymbol{t}_1'}$ w.r.t. common orthogonalizations including Gram-Schmidt-based and SVD-based in Appendix A.2. Under two conditions, we can imagine a case where $\boldsymbol{r}_1'$ are far away from its ground truth $\boldsymbol{r}_1$ but the gradient on $\boldsymbol{t}_1'$ is zero because of the influence from the second and third column. This will greatly lower down the training speed.

## A.4 PROOF OF THEOREM 1

*Proof.* Let $i \in \{0, 1\}$. By the chain rule,

$$\frac{\partial L(\hat{\mathbf{R}}_i)}{\partial \boldsymbol{w}} = \frac{\partial L(\hat{\mathbf{R}}_i)}{\partial \hat{\mathbf{R}}_i} \frac{\partial \hat{\mathbf{R}}_i}{\partial \hat{\mathbf{R}}_0} \frac{\partial \hat{\mathbf{R}}_0}{\partial \boldsymbol{w}}. \tag{17}$$

By the definition of $\epsilon$,

$$\epsilon^2 \geq \left\| \left( \frac{\partial L(\hat{\mathbf{R}}_i)}{\partial \boldsymbol{w}} \right)^{\intercal} \right\|^2 = \left\| \left( \frac{\partial L(\hat{\mathbf{R}}_i)}{\partial \hat{\mathbf{R}}_i} \frac{\partial \hat{\mathbf{R}}_i}{\partial \hat{\mathbf{R}}_0} \frac{\partial \hat{\mathbf{R}}_0}{\partial \boldsymbol{w}} \right)^{\intercal} \right\|^2 \tag{18}$$

$$= \frac{\partial L(\hat{\mathbf{R}}_i)}{\partial \hat{\mathbf{R}}_i} \frac{\partial \hat{\mathbf{R}}_i}{\partial \hat{\mathbf{R}}_0} \left( \frac{\partial \hat{\mathbf{R}}_0}{\partial \boldsymbol{w}} \left( \frac{\partial \hat{\mathbf{R}}_0}{\partial \boldsymbol{w}} \right)^{\intercal} \right) \left( \frac{\partial \hat{\mathbf{R}}_i}{\partial \hat{\mathbf{R}}_0} \right)^{\intercal} \left( \frac{\partial L(\hat{\mathbf{R}}_i)}{\partial \hat{\mathbf{R}}_i} \right)^{\intercal} \tag{19}$$

$$= \frac{\partial L(\hat{\mathbf{R}}_i)}{\partial \hat{\mathbf{R}}_i} \mathbf{B}_i \left( \mathbf{D}\mathbf{D}^{\intercal} \right) \mathbf{B}_i^{\intercal} \left( \frac{\partial L(\hat{\mathbf{R}}_i)}{\partial \hat{\mathbf{R}}_i} \right)^{\intercal} \tag{20}$$

Since $\mathbf{D}\mathbf{D}^{\intercal}$ is a real symmetric matrix, by the eigendecomposition of $\mathbf{D}\mathbf{D}^{\intercal} = \mathbf{Q}\Lambda\mathbf{Q}^{\intercal}$, we have

$$\epsilon^2 \geq \frac{\partial L(\hat{\mathbf{R}}_i)}{\partial \hat{\mathbf{R}}_i} \mathbf{B}_i \left( \mathbf{D}\mathbf{D}^{\intercal} \right) \mathbf{B}_i^{\intercal} \left( \frac{\partial L(\hat{\mathbf{R}}_i)}{\partial \hat{\mathbf{R}}_i} \right)^{\intercal} \tag{21}$$

$$= \left( \frac{\partial L(\hat{\mathbf{R}}_i)}{\partial \hat{\mathbf{R}}_i} \mathbf{B}_i \mathbf{Q} \right) \Lambda \left( \mathbf{Q}^{\intercal} \mathbf{B}_i^{\intercal} \left( \frac{\partial L(\hat{\mathbf{R}}_i)}{\partial \hat{\mathbf{R}}_i} \right)^{\intercal} \right) \tag{22}$$

$$= \sum_k \Lambda_{kk} \left( \mathbf{Q}_k^{\intercal} \mathbf{B}_i^{\intercal} \left( \frac{\partial L(\hat{\mathbf{R}}_i)}{\partial \hat{\mathbf{R}}_i} \right)^{\intercal} \right)^2 \tag{23}$$

$$\geq \lambda_{\min}(\mathbf{D}\mathbf{D}^{\intercal}) \sum_k \left( \mathbf{Q}_k^{\intercal} \mathbf{B}_i^{\intercal} \left( \frac{\partial L(\hat{\mathbf{R}}_i)}{\partial \hat{\mathbf{R}}_i} \right)^{\intercal} \right)^2 \tag{24}$$

$$= \lambda_{\min}(\mathbf{D}\mathbf{D}^{\intercal}) \left\| \mathbf{Q}^{\intercal} \mathbf{B}_i^{\intercal} \left( \frac{\partial L(\hat{\mathbf{R}}_i)}{\partial \hat{\mathbf{R}}_i} \right)^{\intercal} \right\|^2 \tag{25}$$

$$= \lambda_{\min}(\mathbf{D}\mathbf{D}^{\intercal}) \left\| \mathbf{B}_i^{\intercal} \left( \frac{\partial L(\hat{\mathbf{R}}_i)}{\partial \hat{\mathbf{R}}_i} \right)^{\intercal} \right\|^2. \tag{26}$$

By expanding the squared Euclidean norm,

$$\epsilon^2 \geq \lambda_{\min}(\mathbf{D}\mathbf{D}^{\intercal}) \frac{\partial L(\hat{\mathbf{R}}_i)}{\partial \hat{\mathbf{R}}_i} (\mathbf{B}_i \mathbf{B}_i^{\intercal}) \left( \frac{\partial L(\hat{\mathbf{R}}_i)}{\partial \hat{\mathbf{R}}_i} \right)^{\intercal}. \tag{27}$$

Since $\mathbf{B}_i \mathbf{B}_i^{\intercal}$ is a real symmetric matrix, by repeating the same proof steps with eigendecomposition of $\mathbf{B}_i \mathbf{B}_i^{\intercal}$,

$$\epsilon^2 \geq \lambda_{\min}(\mathbf{D}\mathbf{D}^{\intercal}) \lambda_{\min}(\mathbf{B}_i \mathbf{B}_i^{\intercal}) \left\| \left( \frac{\partial L(\hat{\mathbf{R}}_i)}{\partial \hat{\mathbf{R}}_i} \right)^{\intercal} \right\|^2. \tag{28}$$

| | MPJPE ↓ | PA-MPJPE ↓ | MPVPE ↓ |
|---|---|---|---|
| HMR Kanazawa et al. (2018) | 74.4 | 46.6 | 87.3 |
| HMR + PRoM | **71.4** | **44.5** | **84.6** |
| PARE Kocabas et al. (2021) | 82.9 | 52.3 | 99.7 |
| PARE + PRoM | **80.5** | **49.7** | 96.8 |
| CLIFF (Res50) Li et al. (2022) | 72.0 | 45.7 | 85.3 |
| CLIFF (Res50) + PRoM | **70.8** | **44.5** | **84.1** |

Table 4: Evaluation of different models on 3DPW. '+PRoM' means replacing the 6D representation with a Pseudo Rotation Matrix representation.

If $\lambda_{\min}(\mathbf{D}\mathbf{D}^\intercal)\lambda_{\min}(\mathbf{B}_i\mathbf{B}_i)^\intercal \neq 0$, by taking the square root of both sides, this implies that

$$\left\| \left( \frac{\partial L(\hat{\mathbf{R}}_i)}{\partial \hat{\mathbf{R}}_i} \right)^\intercal \right\| \leq \epsilon \psi(\mathbf{D})\psi(\mathbf{B}_i). \tag{29}$$

Since the training loss $\boldsymbol{w} \to L(\hat{\mathbf{R}}_i)$ is non-convex while the loss criterion $\hat{\mathbf{R}}_i \to L(\hat{\mathbf{R}}_i)$ is convex, we have that

$$L(\mathbf{R}^*) \geq L(\hat{\mathbf{R}}_i) + \frac{\partial L(\hat{\mathbf{R}}_i)}{\partial \hat{\mathbf{R}}_i}(\mathbf{R}^* - \hat{\mathbf{R}}_i). \tag{30}$$

This implies that

$$L(\hat{\mathbf{R}}_i) \leq L(\mathbf{R}^*) + \frac{\partial L(\hat{\mathbf{R}}_i)}{\partial \hat{\mathbf{R}}_i}(\hat{\mathbf{R}}_i - \mathbf{R}^*) \tag{31}$$

$$\leq L(\mathbf{R}^*) + \left\| \left( \frac{\partial L(\hat{\mathbf{R}}_i)}{\partial \hat{\mathbf{R}}_i} \right)^\intercal \right\| \|\hat{\mathbf{R}}_i - \mathbf{R}^*\| \tag{32}$$

$$\leq L(\mathbf{R}^*) + \epsilon E \psi(\mathbf{D})\psi(\mathbf{B}_i). \tag{33}$$

### A.5 Proof of Theorem 2

*Proof.* Since $\mathbf{B}_0 = \frac{\partial \hat{\mathbf{R}}_0}{\partial \hat{\mathbf{R}}_0} = \mathbf{I}$, we have $\psi(\mathbf{B}_0) = 1/\sqrt{\lambda_{\min}(\mathbf{B}_0\mathbf{B}_0^\intercal)} = 1/\sqrt{\lambda_{\min}(I)} = 1$. This proves the statement for $\psi(\mathbf{B}_0)$. For the statement of $\psi(\mathbf{B}_1)$, we invoke a part of the inverse function theorem: i.e., if the determinant of the Jacobian matrix of $g$ at its current input $\hat{\mathbf{R}}_0$ is nonzero, then $g$ is locally injective at $\hat{\mathbf{R}}_0$. This implies that if $g$ is not locally injective at $\hat{\mathbf{R}}_0$, then the determinant of the Jacobian matrix of $g$ at $\hat{\mathbf{R}}_0$ is zero. By noticing that the Jacobian of $g$ is $\mathbf{B}_1$, this implies that $\lambda_{\min}(\mathbf{B}_1\mathbf{B}_1^\intercal) = 0$ and hence $\psi(\mathbf{B}_1) = \infty$.

### A.6 On non-injectivity of orthonormalization

As a simple illustration, let $g(\mathbf{A})$ represent a Gram–Schmidt orthonormalization of a given matrix $\mathbf{A} = [v_1, v_2, \ldots, v_n]$. Then, we can always perturb the second column of $\mathbf{A}$ by $\mathbf{A} + \epsilon\Delta$ such that $g(\mathbf{A}) = g(\mathbf{A} + \epsilon\Delta)$ for any sufficiently small $\epsilon$ by setting $\Delta = [0, \delta, 0, \ldots, 0]$ for any $\delta$ satisfying $\delta^\intercal v_1 = 0$. Thus, an orthogonalization map $g$ is inherently not locally injective and it is shown to cause the issue for convergence to optimal matrices in Theorems 1–2.

### A.7 Proof for downstream tasks

Similar to the above proof, we can arrive at

$$L(\hat{\mathbf{Y}}_i) \leq L(\mathbf{Y}^*) + \epsilon E \psi(\mathbf{D})\psi(\mathbf{W})\psi(\mathbf{B}_i), \text{ where } \mathbf{W} = \frac{\partial \hat{\mathbf{Y}}}{\partial \hat{\mathbf{R}}_i}. \tag{34}$$

Since the downstream tasks usually consist of linear operations, it's easy to obtain that $\psi(\mathbf{W}) \neq 0$.

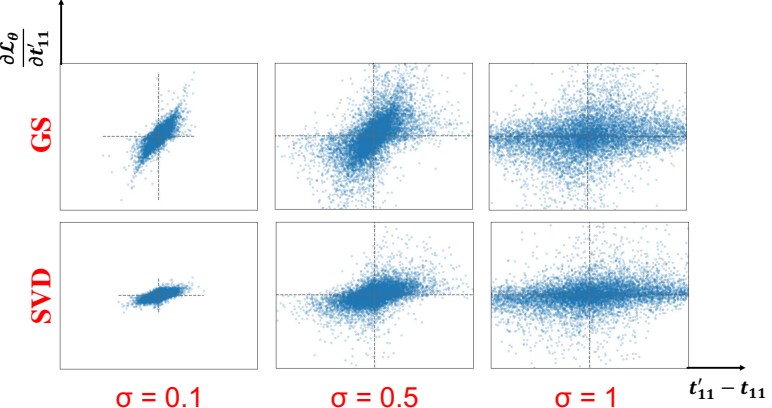

Figure 3: The ambiguity of incorporating Gram-Schmidt (GS) and SVD exists under each $\sigma$. With a larger $\sigma$, the gradient for the same $x$ becomes more diverse which indicates that at the beginning stage of training, the ambiguous gradients are severe and we claim this will influence the training efficiency.

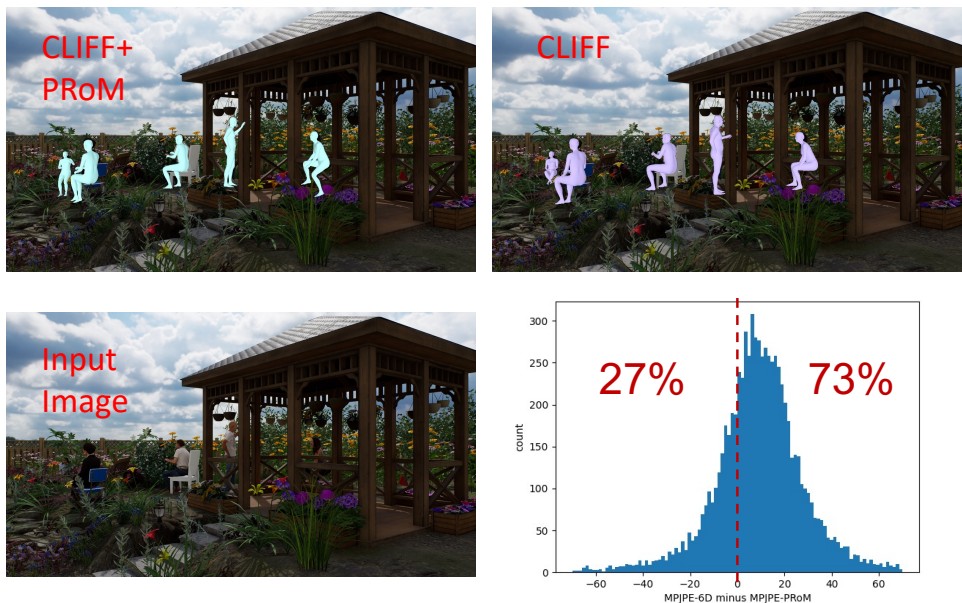

Figure 4: Qualitative results of PRoM vs 6D-based rotation representation on AGORA. We can see that PRoM achieves general improvement in samples. Statistically, with the same backbone, PRoM outperforms 6D-based representation on over 70% of the total samples.

## A.8 UPDATE AMBIGUITY WITH DIFFERENT $\sigma$S

To fully understand the update ambiguity under different noises, we present the visualization under three different $\sigma$s: 0.1, 0.5, and 1 as shown in Fig. 3. We can see that with the increase in variation, the gradients become more diverse.

## A.9 DETAILS OF EXPERIMENTAL SETTINGS

**Point Cloud Pose Estimation.** For the task of point cloud pose estimation, different rotation representations are estimated to rotate the reference point cloud $\mathbf{P}_r$ to a target point cloud $\mathbf{P}_t$. The network is required to directly output estimated rotations. During training, the loss for the point cloud pose estimation task $\mathcal{L}_{\text{pc}}$ is

$$\mathcal{L}_{\text{pc}} = \mathcal{L}_\theta + \mathcal{L}_{\text{ele}}(\hat{\mathbf{P}}_t, \mathbf{P}_t). \tag{35}$$

Different rotation representations have different $\hat{\theta}$s and $\theta$s. For axis-angle and Euler angle, $\theta \in \mathbb{R}^3$; for quaternions, $\theta \in \mathbb{R}^4$; for 6D representations, $\mathcal{L}_\theta = \mathcal{L}_{\text{ele}}(f_{\text{GS}}(\theta), \mathbf{R})$ where $\theta \in \mathbb{R}^6$; for the proposed PRoM, $\theta \in \mathbb{R}^9$.

In summary, point cloud pose estimation is a task in which both rotation loss and downstream loss exist.

**3D Body/Hand Pose and Shape Estimation.** As illustrated in Sec. A.1, there are two downstream outputs, $\hat{\mathbf{Y}}_M$ and $\hat{\mathbf{Y}}_J$, which are directly associated with rotations. Since the previous paper (Kolotouros et al., 2019) has shown the advantage of applying 6D representation over axis-angle. We only compare the proposed PRoM with 6D-representation-based methods. Therefore, the total loss for the task of body/hand pose and shape estimation is

$$\mathcal{L}_{\text{ps}} = \mathcal{L}_\theta + \mathcal{L}_{\text{ele}}(\hat{\mathbf{Y}}_M, \mathbf{Y}_M) + +\mathcal{L}_{\text{ele}}(\hat{\mathbf{Y}}_J, \mathbf{Y}_J). \tag{36}$$

The above loss is applicable in Table 1, 3 (a), 3 (b) and 4. In summary, these experiments indicate the superiority of our method over 6D representation when both rotation loss and downstream loss exist.

**Pose and Shape Estimation with 2D Ground Truth.** In this setting, we have ground truth 2D keypoints as a weak supervision to refine the 3D predictions. Formally, by reprojecting the predicted 3D keypoints $\hat{\mathbf{Y}}_J$ with the estimated camera, we apply an alignment loss between the reprojected 2D prediction and ground truth 2D locations:

$$\mathcal{L}_{\text{ops}} = \mathcal{L}_{\text{ele}}(\hat{\mathbf{Y}}_{J_{2D}}, \mathbf{Y}_{J_{2D}}) = \mathcal{L}_{\text{ele}}(\text{reproj}(\hat{\mathbf{Y}}_J), \mathbf{Y}_{J_{2D}}). \tag{37}$$

Therefore, in this experiment, we demonstrate that when only downstream loss exists, our method still outperforms 6D representations.

## A.10 IMPLEMENTATION DETAILS OF BODY POSE AND SHAPE ESTIMATION TASK

To compare with the state-of-the-art methods on 3D human body and hand mesh recovery, we take the recently introduced CLIFF Li et al. (2022) as the baseline and replace the 6D representation with our proposed pseudo rotation matrices (PRoM). 'CLIFF + PRoM' predicts in total $J \times 9$ pose parameters, where $J$ is the joint number and each joint has a 9D prediction of PRoM. During training, PRoM is used to calculate the pose loss and is fed to the parametric model (SMPL Loper et al. (2015) for body and MANO Romero et al. (2017) for hand) to calculate the 3D joint loss and 2D reprojection loss, instead of the rotation matrices from the Gram-Schmidt process. The training setting is the same as CLIFF, except for the initial learning rate of 3e-4, which will cause gradient explosion in the original 6D version. For the experiments on AGORA Patel et al. (2021), we use ViTPose Xu et al. (2022) as the backbone. More specifically, for body pose, we use the Adam optimizer to train the model for 244K steps with a batch size of 256. The learning rate is reduced by a factor of 10 at the 122K$^{\text{th}}$ step. The input images are cropped using the ground-truth bounding boxes, and resized to $256 \times 192$, preserving the aspect ratio. For hand pose, we resize the cropped images to $224 \times 224$, train the model for 101K steps with a batch size of 128, and reduce the learning rate by a factor of 10 at the 70K$^{\text{th}}$ and 90K$^{\text{th}}$ steps. During inference, we adopt Unbiased Gram-Schimidt process on PRoM to obtain the final rotation matrices.

## A.11 Visualizations of PRoM vs 6D-based methods

We give visualizations using the same backbone but with PRoM and 6D representation on the challenging dataset AGORA in Fig. 4. From the bottom-right figure, we can conclude that PRoM has a general improvement over all samples than 6D-based methods.

## A.12 Details about Ablation of Different Models

We demonstrate the ablation results in Table 4, which show significant improvement over all models. For the implementation details, we adopt the SOTA experiment setting to train the models with a mixture of 3D datasets (Human3.6M Ionescu et al. (2013), MPI-INF-3DHP Mehta et al. (2017) and 3DPW Von Marcard et al. (2018)) and 2D datasets (COCO Lin et al. (2014) and MPII Andriluka et al. (2014)). The 3D pseudo-ground-truth for 2D datasets is provided by the CLIFF annotator Li et al. (2022), except that we use the EFT pseudo-GT Joo et al. (2021) for PARE-based models. The image encoders are based on ResNet-50 He et al. (2016).

