# OpenReview forum: "Learning Unorthogonalized Matrices for Rotation Estimation"
_ICLR.cc/2024/Conference — Submitted to ICLR 2024_

### Official Review · Reviewer_v4Yx · 2023-10-29

**Soundness:** 3 good
**Presentation:** 3 good
**Contribution:** 2 fair
**Rating:** 5
**Confidence:** 4

**Summary:**

The paper presents PRoM, a method to learn pseudo-rotation matrices. The motivation is that the paper claims that learning rotation matrices in a deep learning framework suffers from imposing orthonormality constraints by means of SVD or Gram-Schmidt. The proposed method removes these orthogonalization methods and apply the estimated rotation matrix. The paper presents a mathematical analysis discussing the reasons about why SVD or Gram-Schmidt suffer when applied in a deep learning framework. The paper presents experiments on various applications (e.g., human pose estimation and point cloud pose estimation) showing that PRoM outperforms the included baselines.

**Strengths:**

S1. I think the paper does a good job analyzing the problems that orthogonalization methods introduce when used in the learning frameworks.

S2. I find the experiments interesting because they show the improvements PRoM can bring to human pose estimation, cloud pose estimation, among other applications.

S3. The paper is well written and is easy to follow  and understand. Clarity thus is high and should be easy to replicate.

**Weaknesses:**

While I think the paper touches a very interesting topic (i.e., rotation estimation in deep learning frameworks) and shows interesting results, I have several concerns that make me a bit skeptical about the approach:

1. Lack of discussion about other possible ways to constrain orthonormality in learning problems. While the problems of orthogonalization in the learning problem may bring issues as described in the paper, there are other possible solutions that the paper does not discuss. For example:

i) In 3D computer vision, some camera-pose estimation solvers constrain the rotation with $I - M^{\intercal}M$ and $\text{det}(M)=1$ pose problems and showing good estimation results (see Reference [A]). Why couldn't this trick replace the SVD or Gram-Schmidt orthogonalization procedures? Although, the paper does not discuss a learning approach, the trick is applicable to any learning problem. Unfortunately, the paper lacks discussion about these tricks and if they also present problems when learning.

ii) Learning rotation with neural networks does not need to follow the pipeline shown in Figure 1. One can easily use a quaternion parametrization, remove $\mathcal{L}_{\theta}$ and thus remove the orthogonalization, and simply add a regularizer term that enforces the norm of the quaternion be one.

2. The continuity issues discussed in the submission and that are used to motivate the problem are questionable. One of the main claims of the paper in Learning for Rotations in Section 2 states that the parametrization of 3D rotations with four or fewer dimensions is discontinuous and non-ideal for learning (according to Zhou et al. 2019). However, Zhou et al. 2019 defines the continuity concept from an intuitive example and lacks solid theorems showing that these definitions and observations strictly apply to learning rotations. Subsequent works using quaternions show that they work well with neural networks (see Reference [B] below). Reference B conflicts with the observations of Zhou et al. which is the foundation of this work and explore pseudo-rotation matrices. Thus, this puts into question the main assumptions stated in Section 2.

3. The paper would've been more solid if they included the discussed methods in 1 (see above).

References.
[A] Ventura, Jonathan, et al. "A minimal solution to the generalized pose-and-scale problem." CVPR. 2014.
[B] Zhang et al. Quaternion Product Units for Deep Learning on 3D Rotation Groups. CVPR 2020

**Questions:**

Given the rationale described in the Weaknesses above, I don't find the proposed method of using pseudo-rotation matrices that convincing. It marries to the pipeline of learning rotations to that one described in Figure 1 of the submission. However, I think there are other ways to learn rotations that are not discussed in the submission.

---

> ### Author Response · Authors · 2023-11-23
> **Thank you for your review**
>
> Thanks for providing feedback on our submission. We address individual concerns below.
>
>
> **Q1: Contrain orthonormality with $I-M^{\intercal}M$?**
>
> A1: Thank you for your suggestion. We compare our method with adding the rotation constraint loss, which is implemented by ProHMR [1]. The PA-MPJPE is 55.3 vs. 54.8 (PRoM). The experimental setting is the same as the ablation study. Adding this loss makes the learning in between PRoM and previous orthogonalization incorporated methods. The results are also in between.
>
> [1] Kolotouros, Nikos, et al. Probabilistic modeling for human mesh recovery. CVPR'21
>
> &nbsp;
>
>
> **Q2: Quaternions are also used in other works; Continuity lacks solid theorems**
>
> A2: In our work, we focus on whether to use orthogonalizations during training with the matrix representation. 6D is one of the representative works that incorporate orthogonalizations. 6D is also frequently used in pose estimation tasks. Our work mainly improves upon 6D representation, which will have an impact on the understanding of learning rotation matrices and the further learning paradigm of pose estimation. We acknowledge that other rotation representations, e.g., quaternions, are used in other works, comparing 6D with quaternion with solid theorems is out of the scope of our paper.
>
> However, we still provide a more detailed empirical comparison among all rotation representations. The experimental setting is the same as the ablation studies, e.g., Table 3 (a). The results are shown below.
>
> | Representation | Quaternion | axis-angle | Euler | 6D   | SVD  | PRoM |
> |----------------|------|------------|-------|------|------|------|
> | PA-MPJPE       | 62.8 | 64.3       | 66.8  | 57.5 | 56.7 | **54.8** |
>
> **Q3: include the discussed methods**
>
> A3: Thank you for your suggestions. We will include all the experiments above in our manuscript.

---

### Official Review · Reviewer_drye · 2023-10-29

**Soundness:** 3 good
**Presentation:** 2 fair
**Contribution:** 2 fair
**Rating:** 6
**Confidence:** 3

**Summary:**

The work studied the rotation representation for rotation estimation and other downstream tasks, such as human body and hand shape/pose estimation. An issue with orthogonalization in learning was pointed out: this could lead to explosive gradient and hence harm the training stage stability. To overcome this, this work proposes to get rid of the step of orthogonalization during learning, and only including this step during inference step. Theoretical proofs are derived to support the claim of the proofs and evaluations on multiple tasks verify the effectiveness of the proposed rotation representation.

**Strengths:**

1. The work focus on the choice of rotation representation, the very basic but vital ingredient in rotation estimation task. This is in contrast to most work in 3D pose estimation, and has the potential of larger impact in the field.
2. The method is simple yet effective. The motivation for the method is put clear. Gradient update is affected by the extra step of orthogonalization. By simply removing it, the baseline method could be improved.
3. Evaluations tasks are very extensive. These include human body, hand, and pure rotation estimation tasks. Three cases are considered with respect to the availability of intermediate rotation ground truth / end-task supervision.

**Weaknesses:**

1. The evaluations consider two baselines (Zhou et al., 2019; Levinson et al., 2020). Both methods use 6D representations with differences in normalization. A comparison to more recent methods [1,2] should be more convincing.

2. The proposed method chose to base on CLIFF (Li et al., 2022). Since the proposed method is not a novel framework, it is generally applicable to any existing methods for pose estimation. Hence, I assume that PROM could be used together with baselines to also improve learning.

3. In the area of camera pose regression, it has been shown that predicting pseudo rotation matrix during training works in [3] and quantanion without orthogonalization is also possible [4]. As a result, the novelty of this work might be reduced to introducing the simple idea to other tasks.


[1] Projective Manifold Gradient Layer for Deep Rotation Regression, Jiayi Chen etc.
[2] Deep Projective Rotation Estimation through Relative Supervision, Brian Okorn etc.
[3] Direct-PoseNet: Absolute Pose Regression with Photometric Consistency, Shuai Chen, etc.
[4] Posenet: A convolutional network for real-time 6-dof camera relocalization. Alex Kendall, etc.

**Questions:**

1. There may be a typo in Sec 3.4.1. In Eq.5, the gradient is respect to $t'_1$, but the text refers to $t'_2$ and $t'_3$ to show the column-gradient will be in different directions. Please confirm this.
2. This method seems to be very simple, simply removing the $r$ and $g$. why is removing $r$ possible? If I understand correct, $r$ is the mapping from input to rotation representation, and should not be identity mapping.
3. My biggest concern is: if orthogonalization can be safely removed as suggested by this work, why the previous work adopted GS or SVD to make the rotation representation valid?

---

> ### Author Response · Authors · 2023-11-23
> **Thank you for your review**
>
> Thanks for providing feedback on our submission. We address individual concerns below.
>
> **Q1: Lack of comparison with recent methods**
>
> A1: Thank you for pointing this out. Considering [B] didn’t release their code, we reproduce the results of [A] with our ablation study setting. With the only change of rotation representation from 6D to RPMG-6D, [A] achieves the PA-MPJPE of  55.8, whereas PRoM has a lower error of 54.8. We want to highlight that one of the main contributions of our paper is that we achieve much better results on the challenging real-world human pose datasets, e.g., 3DPW and Human3.6M.
>
> **Q2: PRoM is applicable to other baselines in pose estimation?**
>
> A2: Yes, PRoM will improve other baselines. We have already included these experiments in the last paragraph of Sec. 5.3. The detailed numbers are presented in Table 4 in the Appendix due to the space limit.
>
> **Q3: it has been shown that predicting pseudo rotation matrix and quaternion without orthogonalization is also possible**
>
>
> A3: Yes, also as reviewer h3do commented, direct regression of rotation matrices is intuitive and has been used in the past literature. However, our contribution is mainly the gradient and optimization analysis, which shows that incorporating orthogonalization in the training is problematic. Our work gives theoretical support for why using unorthogonalized matrices is more beneficial.

---

### Official Review · Reviewer_h3do · 2023-11-01

**Soundness:** 1 poor
**Presentation:** 2 fair
**Contribution:** 1 poor
**Rating:** 3
**Confidence:** 5

**Summary:**

This paper claims that a popular rotation estimation technique (6D representation) for deep learning has a critical flaw, i.e., ambiguous gradient. The paper tries to demonstrate this with some derivations, and the authors propose to use the pseudo rotation matrix technique (using a plain 3x3 matrix directly, with an additional loss that makes it similar to the ground truth rotation) to overcome this issue. This pseudo rotation matrix is orthogonalized in the test phase. Experiments show that the proposed method has less variation in the gradient, and it shows somewhat better performance in several downstream tasks, including 3D body/hand pose and shape estimation.

**Strengths:**

The topic of the paper (rotation estimation in deep learning) has some importance in the field.

**Weaknesses:**

- To tell the truth, I was one of the reviewers of this paper in a previous venue. I see that the authors have removed some wrong derivations, but there are still many vague and wrong claims in the paper.

- Most claims in the paper are either vague or not surprising (not showing what the authors intended to show). Most importantly, the proposed method (using a plain 3x3 matrix without any orthogonalization and instead guiding it to a rotation matrix using an additional loss, i.e., soft constraint) is a somewhat basic technique that has been frequently used in the literature. Accordingly, I'm afraid the paper has little merit to the research community. The detailed errors (or concerns I have) are listed below:

- Section 3.4.1 (and Appendix A.3) basically says that (5) has two more terms other than the first term, so the gradient can be scrambled. The claim itself is vague, but before that, what the authors need to show is not that there are two more terms, but that these two terms do not die out (and actually steer the overall gradient towards funny directions) in the majority of the input space. In (12)-(16), we can actually see that there are many duplicating expressions, as well as many terms related to cross-product, inside those three terms. This suggests that the two terms (related to $r_2$ and $r_3$) might actually be canceled, either by themselves or by combining the two. I actually showed this in my previous review for a specific case provided in the previous manuscript. This might hold for larger regions in the input space if we do the manipulation. The authors must explicitly derive against this to justify their claim, not suggesting them as in the current paper.

- Theorems in Section 4.2 are not surprising, and most importantly, they do not prove anything. What they are showing are (specific) upper bounds of losses for orthogonalized and unorthogonalized cases. This does not guarantee anything. They do not guarantee that the actual loss will be lower for the unorthogonalized case, and moreover, there is no proof that they are the best upper bounds we can have (so it is not correct to say that the unorthogonalized case has a better chance of a lower loss).

- The meaning of Fig. 3 is vague. It simply shows that the correlation between the gradient and $t_{11}$ is not apparent in Euclidean space. This has nothing to do with the stability or ambiguity of the gradient. More importantly, it is obvious that the relation between the gradient and the parameter is more complex for orthogonalization than for plain identity mapping.

- What if we do not have ground truth rotations? Without $L_\theta$, it is possible that PRoM deviates too far from the rotation space, and this can be detrimental to having a correct solution. What should we do then?

**Questions:**

Please see the above weaknesses.

---

> ### Author Response · Authors · 2023-11-23
> **Thank you for your review**
>
> Thanks for providing feedback on our submission. We address individual concerns below.
>
> **Q1: The proposed method is somewhat basic and is frequently used in the literature.**
>
> A1: PRoM is just a simple yet effective solution according to our optimization analysis, which is the main contribution of our paper. Most works still incorporate orthogonalizations, e.g., in the fields of point cloud pose estimation, human pose and shape estimation tasks. People tend to apply orthogonalizations on rotation matrices to ensure it’s valid to rotate the downstream inputs. It’s a tradeoff between the validity of rotation matrices and optimization. Our results show that the optimization matters more.
>
> **Q2: Eq. (5) is vague and need to explicitly derive the two terms will not die out?
> Q4: Fig. 3 is vague? Incorporating orthogonalization is obviously more complex.**
>
> A2 & A4: We answer the concerns together since they are related. For the ambiguous gradient, we only give some intuitive derivation (Eq. (5)) and Fig. 2 (a) is the exact empirical verification. Fig. 3 serves as a supplement to Fig. 2 (a). We leave explicit theoretical support for this in Sec. 4.2. We answer the questions separately below in detail.
>
> A2: Eq. (5) is an observation of the gradients from the orthogonalizations and serves as the motivation of PRoM. The explicit influence of these conflicting terms on the optimization is shown in Sec. 4.2. We leave strong theoretical proof for the optimization analysis instead of gradient because the optimization analysis is more established in the convergence and optimization papers (see references below) Therefore, we think it’s more convincing to demonstrate the negative impact of the orthogonalizations in the optimization analysis. Also, the two terms will not die out as shown in Fig. 2 (a).
>
> A4: Fig. 3 is an empirical study with different $\sigma$s for comprehensive understanding. The focus is that the larger $\sigma$ is, the more diversity of gradients will be. It serves as a supplement of Fig. 2 (a). As stated in the review, it is indeed obvious that the gradient is more complex for orthogonalization. We are actually targeting the complex gradient and demonstrating both the theoretical and empirical benefits of removing this complex gradient during the training.
>
> **Q3: better upper bound does not guarantee anything?**
>
> A3: The upper bound analysis of loss is widely adopted both in machine learning theory and its application in vision [1,2,3,4,5,6]. It both considers the convergence rate and irreducible error. Our work follows the convention of analyzing the different upper bounds for orthogonalization incorporated methods and PRoM. For the whole term, the only difference is caused by the orthogonalization function, which gives non-negligible $\psi(B_i)$ and further leads to a slower convergence rate and worse upper bound. Additionally, we show by experiments that PRoM converges much faster than orthogonalization incorporated methods (Fig. 3 (a) and (b)).
>
> [1] Cohen, Jeremy M., et al. Gradient Descent on Neural Networks Typically Occurs at the Edge of Stability. ICLR’21
>
> [2] Ahn, Kwangjun, Jingzhao Zhang, and Suvrit Sra. Understanding the unstable convergence of gradient descent. ICML’22
>
> [3] Fehrman, Benjamin, Benjamin Gess, and Arnulf Jentzen. Convergence rates for the stochastic gradient descent method for non-convex objective functions. JMLR’20
>
> [4] Mertikopoulos, Panayotis, et al. On the almost sure convergence of stochastic gradient descent in non-convex problems. Neurips’20
>
> [5] Lei, Yunwen, et al. Stochastic gradient descent for nonconvex learning without bounded gradient assumptions. TNNLS’22
>
> [6] Chen, Joya, et al. DropIT: Dropping Intermediate Tensors for Memory-Efficient DNN Training. ICLR’23
>
> **Q5: What if we do not have ground truth rotations?**
>
> A5: Our analysis still applies when ground truth rotations are not available. In Theorem 1, we do not need the presence of the ground truth $R$, but only the predicted rotation matrices $\hat{R}$. Also, empirically, in sec. 5.2.3, we show that when the whole task only has the downstream supervision without ground truth rotation, it can still perform better than orthogonalization incorporated methods.

---

> ### Comment · Reviewer_h3do · 2023-12-04
> **I'm not convinced.**
>
> The authors have not answered my concerns regarding whether the two of the three terms cancel each other. This is the core argument in the paper, and I have already demonstrated the case where they cancel each other in the previous round.
>
> If there is no ground truth rotation, then there will be no term that makes the (pseudo-) rotation estimate close to a rotation matrix. Of course, this might be easier for optimization, but the estimate can be far from a rotation matrix, which can cause the quality of the solution to be poor.
>
> I'm not at all convinced by the upper-bound argument. The papers the authors mentioned show upper bounds to guarantee the least bound for convergence for some specific scenarios. However, comparing upper bounds cannot prove that orthogonalization is worse than no-orthogonalization.

---

### Meta-Review · Area_Chair_Pi6V · 2023-12-05

**Metareview:**

In the context of the estimation or production of rotation matrices from machine learning systems, it is common for learning systems to incorporate the constraint that the rotation matrices should be orthonormal, as is required for almost all downstream tasks.  As the reviewers observe, there are several standard approaches - those mentioned by the reviewers are here, and there are others:
 1. Ensure orthonormality by taking an unconstrained 3x3 matrix and orthonormalizing via Gram Schmidt or SVD
 2. Ensure orthonormality by taking a quaternion, normalizing, and converting to rotation
 3. Encourage orthonormality by adding a loss such as norm(R*R' - I)
 4. Encourage orthonormality by taking an unnormalized quaternion, and adding a loss such as abs(q'*q - 1)
 5. Do not attempt to impose orthnormality

The paper considers options 1 and 5, and presents some theorems and experiments to suggest that 5 may be preferred to 1.  As the reviewers note, it is not unusual to attempt 5, but the paper's contribution is to present some arguments as to why 5 may work better than a simple analysis might suggest.

The authors claim to have addressed reviewer h3do's questions in this manuscript, but the reviewer disagrees, and makes a convincing case that the questions remain unaddressed.  The reviewer's argument that upper bounds cannot be compared is true.  The rebuttal avoids the reviewer's questions rather than addressing them.

Reviewer v4Yx also points to the need to investigate and compare to methods 2-4.  The rebuttal presents some results on (3), showing it is very close to PRoM, with no details on how any hyperparameters were tuned, for example weighting factors in the loss.   This result alone argues for resubmission of the manuscript with these experiments much more thoroughly fleshed out.

The authors appear to suggest that it is improper for reviewer h3do to refer to their previous review of the manuscript, but I find no evidence that the reviewer has not searched the current manuscript for answers to their questions.  The very fact that the rebuttal does not clearly provide those answers further evidences this point.

**Justification For Why Not Higher Score:**

The analyses may be valuable, but by not including very natural alternatives, the manuscript will not advance scholarship in the area.  When those analyses are included, it will be valuable to publish.   The experiments added in rebuttal need to be much more thoroughly incorporated for the next review.

**Justification For Why Not Lower Score:**

n/a

---

### Decision · Program_Chairs · 2024-01-16

Reject